# Expecting The Unexpected: Towards Broad Out-Of-Distribution Detection

**Charles Guille-Escuret**
ServiceNow Research, Mila,
Université de Montréal

**Pierre-André Noël**
ServiceNow Research

**Ioannis Mitliagkas**
Mila, Université de Montréal,
Canada CIFAR AI chair,
Archimedes Unit, Athena Research Center

**David Vazquez**
ServiceNow Research

**Joao Monteiro**
Autodesk[2]

## Abstract

Deployed machine learning systems require some mechanism to detect out-of-distribution (OOD) inputs. Existing research mainly focuses on one type of distribution shift: detecting samples from novel classes, absent from the training set. However, real-world systems encounter a broad variety of anomalous inputs, and the OOD literature neglects this diversity. This work categorizes five distinct types of distribution shifts and critically evaluates the performance of recent OOD detection methods on each of them. We publicly release our benchmark under the name BROAD (Benchmarking Resilience Over Anomaly Diversity). We find that while these methods excel in detecting novel classes, their performances are inconsistent across other types of distribution shifts. In other words, they can only reliably detect unexpected inputs that they have been specifically designed to expect. As a first step toward broad OOD detection, we learn a Gaussian mixture generative model for existing detection scores, enabling an ensemble detection approach that is more consistent and comprehensive for broad OOD detection, with improved performances over existing methods. We release code to build BROAD to facilitate a more comprehensive evaluation of novel OOD detectors.[1].

## 1 Introduction

A significant challenge in deploying modern machine learning systems in real-world scenarios is effectively handling out-of-distribution (OOD) inputs. Models are typically trained in closed-world settings with consistent data distributions, but they inevitably encounter unexpected samples when deployed in real-world environments. This can both degrade user experience and potentially result in severe consequences in safety-critical applications [41, 72].

There are two primary approaches to enhancing the reliability of deployed systems: OOD robustness, which aims to improve model accuracy on shifted data distributions [18, 21], and OOD detection [84, 13], which seeks to identify potentially problematic inputs and enable appropriate actions (e.g., requesting human intervention).

Robustness is often considered preferable since the system can operate with minimal disruption, and has been investigated for various types of distribution shifts [69, 27, 30]. However, attaining

---

[1]BROAD is freely accessible under a Creative Commons Attribution 4.0 Unported License at `https://huggingface.co/datasets/ServiceNow/PartialBROAD`. Code and instructions to build the full dataset are available at `https://github.com/ServiceNow/broad`. We use OpenOOD [85] for evaluations.

[2]Work done while at ServiceNow.

38th Conference on Neural Information Processing Systems (NeurIPS 2024) Track on Datasets and Benchmarks.

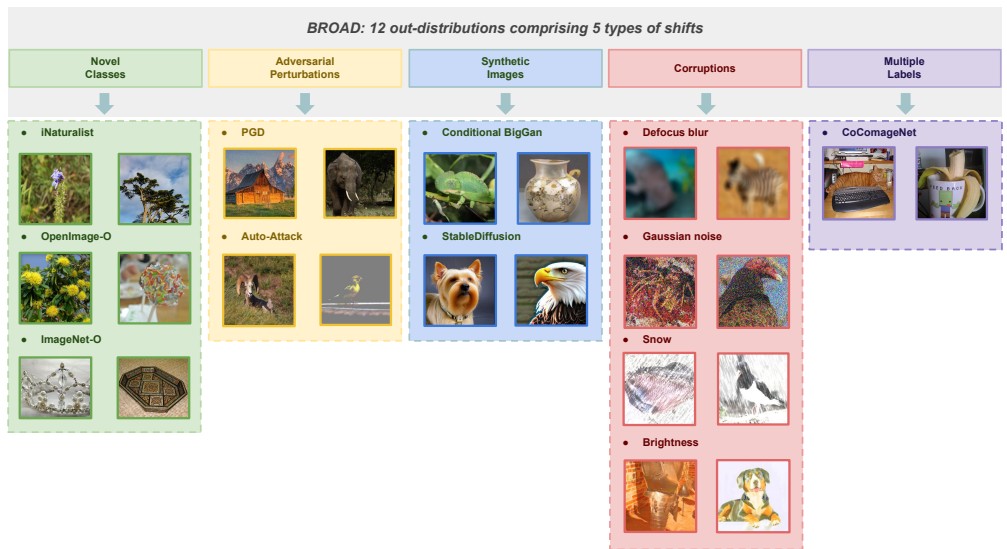

Figure 1: An overview of BROAD: illustrating the benchmarks employed for each distribution shift category, with ImageNet-1K serving as the in-distribution reference.

robustness can be challenging: it may be easier to raise a warning flag than to provide a "correct" answer.

Furthermore, robustness is not achievable when a classification system is presented with an input of an unknown semantic class, as none of the known labels can be considered correct. In recent years, OOD detection research has tackled the detection of such distributions shifts, under different terminologies motivated by subtle variations: open set recognition (OSR), anomaly detection, novelty detection, and outlier detection (see Yang et al. [84] for a comprehensive analysis of their differences).

Beyond novel classes, researchers investigated the detection of adversarial attacks [2, 33] and artificially generated images [36, 53, 51], although these distribution shifts are rarely designated as "OOD". Few works simultaneously detect novel labels and adversarial attacks [45, 25], and the broad detection of diverse types of distribution shifts remains largely unaddressed.

In real-world scenario, ***any* type of distribution shift is susceptible to affect performances and safety**. While recent efforts like OpenOOD [85] have simplified and standardized OOD detection evaluations, their exclusive focus on a specific type of distribution shift is susceptible to yield detection methods that are overspecialized and perform unreliably on *out-of-distribution distribution shifts*, i.e., they only detect "unexpected" samples that are, in fact, expected.

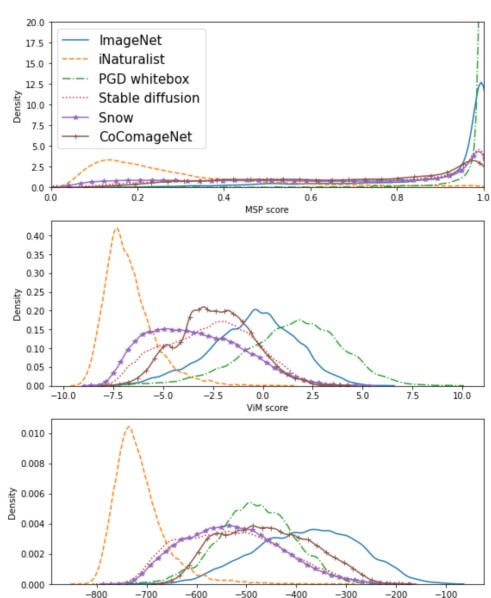

Figure 2: Score distributions of MSP, ViM, and MDS across datasets. While all methods discriminate between ImageNet and iNaturalist, their effectiveness fluctuates across the other types of distribution shifts described in Section 2.

These concerns are confirmed in Figure 2, which displays the distributions of maximum softmax (MSP) [28], ViM [82], and MDS [46] scores on several shifted distributions relative to clean data (ImageNet-1k). Although all scores effectively distinguish samples from iNaturalist [34, 78], a common benchmark for detecting novel classes, their performance on other types of distribution shifts is inconsistent.

Furthermore, OOD detection methods often require tuning or even training on OOD samples [49, 46, 48], exacerbating the problem. Recent research has attempted the more challenging task of performing detection without presuming access to such samples [56, 25, 82]. Nevertheless, they may still be inherently specialized towards specific distribution shifts. For example, CSI [75] amplifies the detection score by the norm of the representations. While this improves performance on samples with novel classes (due to generally lower norm representations), it may conversely impair performance in detecting, for instance, adversarial attacks, which may exhibit abnormally high representation norms.

The scarcity of diversity in OOD detection evaluations in previous studies may be attributed to the perceived preference for OOD robustness when OOD samples share classes with the training set. Nevertheless, this preference may not always be well-founded. Firstly, previous works have indicated a potential trade-off between in-distribution accuracy and OOD robustness [77, 90], although a consensus remains elusive [87]. On the other hand, many OOD detection systems serve as post-processors that do not impact in-distribution performances. Additionally, there are practical scenarios where the detection of OOD inputs proves valuable, regardless of robustness. For instance, given the increasing prevalence of generative models [68, 64, 70], deployed systems may need to differentiate synthetic images from authentic ones, independent of performance [51, 42]. Lastly, other types of shifts exist where labels belong to the training set, but correct classification is undefined, rendering robustness unattainable (see section 2.5).

Our work focuses on *broad OOD detection*, which we define as the simultaneous detection of OOD samples from diverse types of distribution shifts. Our primary contributions include:

- Benchmarking Resilience Over Anomaly Diversity (BROAD), an extensive OOD detection benchmark (relative to ImageNet) comprising twelve datasets from five types of distribution shifts: novel classes, adversarial attacks, synthetic images, corruptions, and multi-class.
- A comprehensive benchmarking of recent OOD detection methods on BROAD.
- The development and evaluation of a generative ensemble method based on a Gaussian mixture of existing detection statistics to achieve broad detection against all types of distribution shifts, resulting in significant gains over existing methods in broad OOD detection.

Section 2 introduces BROAD while Section 3 presents studied methods and our generative ensemble method based on Gaussian mixtures. In Section 4, we evaluate different methods against each distribution shift. Section 5 provides a synopsis of related work, and we conclude in Section 6.

## 2 Distribution Shift Types in BROAD

In this study, we employ ImageNet-1K [15] as our in-distribution. While previous detection studies have frequently used CIFAR [44], SVHN [62], and LSUN [88] as detection benchmarks, recent work has highlighted the limitations of these benchmarks, citing their simplicity, and has called for the exploration of detection in larger-scale settings [29]. Consequently, ImageNet has emerged as the most popular choice for in-distribution.

Our benchmark, BROAD, encompasses five distinct types of distribution shifts, each represented by one to four corresponding datasets, as summarized in Figure 1. This selection, while not exhaustive, is substantially more diverse than traditional benchmarks, and provides a more realistic range of the unexpected inputs that can be plausibly encountered.

### 2.1 Novel Classes

The introduction of novel classes represents the most prevalent type of distribution shift in the study of OOD detection. In this scenario, the test distribution contains samples from classes not present in the training set, rendering accurate prediction unfeasible.

For this particular setting, we employ three widely used benchmarks: iNaturalist [34, 78], ImageNet-O [31], and OpenImage-O [82, 43].

### 2.2 Adversarial Perturbations

Adversarial perturbations are examined using two well-established attack methods: Projected Gradient Descent (PGD)[57] and AutoAttack[12]. Each attack is generated with an $L_\infty$ norm perturbation

budget constrained to $\epsilon = 0.05$, with PGD employing 40 steps. In its default configuration, AutoAttack constitutes four independently computed threat models for each image; from these, we selected the one resulting in the highest confidence misclassification. A summary of the models' predictive performance when subjected to each adversarial scheme can be found in Table 1. The relative detection difficulty of white-box versus black-box attacks remains an open question. Although white-box attacks are anticipated to introduce more pronounced perturbations to the model's feature space, black-box attacks might push the features further away from the in-distribution samples. To elucidate this distinction and provide a more comprehensive understanding of detection performance, we generate two sets of attacks using both PGD and AutoAttack: one targeting a ResNet50 [26] and the other a Vision Transformer (ViT) [19]. Evaluation is performed on both models, thereby ensuring the inclusion of two black-box and two white-box variants for each attack.

Common practice in the field focuses on the detection of successful attacks. However, identifying failed attempts could be advantageous for security reasons. To cater to this possibility, we appraise detection methods in two distinct scenarios: the standard Distribution Shift Detection (DSD), which aims to identify any adversarial perturbation irrespective of model predictions, and Error Detection (ED), which differentiates solely between successfully perturbed samples (those initially correctly predicted by the model but subsequently misclassified following adversarial perturbation) and their corresponding original images.

Table 1: Prediction accuracy of the two evaluated models across the range of perturbation settings examined in our study.

| Model | Clean Acc. | White-box | | Black-box | |
|---|---|---|---|---|---|
| | | PGD | AA | PGD | AA |
| RN50 | 74.2% | 39.3% | 28.2% | 68.0% | 43.3% |
| ViT | 85.3% | 0.4% | 50.8% | 77.1% | 65.8% |

## 2.3 Synthetic Images

This category of distribution shift encompasses images generated by computer algorithms. Given the rapid development of generative models, we anticipate a growing prevalence of such samples. To emulate this shift, we curated two datasets: one derived from a conditional BigGAN model [7], and another inspired by stable diffusion techniques [70].

In the case of BigGAN, we employed publicly available models[2] trained on ImageNet-1k and generated 25 images for each class. For our stable diffusion dataset, we utilized open-source text-conditional image generative models[3]. To generate images reminiscent of the ImageNet dataset, each ImageNet class was queried using the following template:

```
High quality image of a {class_name}.
```

This procedure was repeated 25 times for each class within the ImageNet-1k label set. Given that a single ImageNet class may have multiple descriptive identifiers, we selected one at random each time.

## 2.4 Corruptions

The term *corruptions* refers to images that have undergone a range of perceptual perturbations. To simulate this type of distribution shift, we employ four distinct corruptions from ImageNet-C [27]: defocus blur, Gaussian noise, snow, and brightness. All corruptions were implemented at the maximum intensity (5 out of 5) to simulate challenging scenarios where OOD robustness is difficult, thus highlighting the importance of effective detection. Analogous to the approach taken with adversarial perturbations, we implement two distinct evaluation scenarios: Distribution Shift Detection (DSD), aiming to identify corrupted images irrespective of model predictions, and Error Detection (ED), discriminating between incorrectly classified OOD samples and correctly classified in-distribution samples, thus focusing solely on errors introduced by the distribution shift.

## 2.5 Multiple Labels

In this study, we propose CoComageNet, a new benchmark for a type of distribution shift that, to the best of our knowledge, has not been previously investigated within the context of Out-of-Distribution

---

[2] https://github.com/lukemelas/pytorch-pretrained-gans
[3] https://huggingface.co/stabilityai/stable-diffusion-2

Table 2: Distribution shift detection AUC for Visual Transformer and ResNet-50 across different types of distribution shifts.

| | Novel classes | | Adv. Attacks | | Synthetic | | Corruptions | | Multi-labels | | Average | |
| --- | --- | --- | --- | --- | --- | --- | --- | --- | --- | --- | --- | --- |
| | ViT | RN50 | ViT | RN50 | ViT | RN50 | ViT | RN50 | ViT | RN50 | ViT | RN50 |
| CADET $m_{in}$ | 20.91 | 66.79 | 67.12 | 62.4 | 59.82 | 55.65 | 79.67 | 87.15 | 54.24 | 56.88 | 56.35 | 65.77 |
| ODIN | 91.73 | 73.58 | 52.29 | 54.44 | 62.74 | 61.49 | 79.68 | 88.52 | 70.75 | 64.46 | 71.44 | 68.5 |
| MAX LOGITS | 95.25 | 73.67 | 59.73 | 59.62 | 66.08 | 57.65 | 83.60 | 90.87 | 71.63 | 62.79 | 75.26 | 68.92 |
| LOGITS NORM | 51.93 | 52.62 | 37.39 | 51.82 | 38.25 | 59.47 | 39.99 | 82.81 | 36.32 | 48.05 | 40.78 | 58.95 |
| MSP | 90.56 | 67.25 | 58.45 | 61.17 | 64.78 | 55.59 | 78.62 | 86.71 | 71.93 | **67.52** | 72.87 | 67.65 |
| MDS$_f$ | 53.35 | 63.52 | 67.73 | 55.04 | 54.92 | 56.18 | 31.47 | 76.52 | 63.43 | 36.81 | 54.18 | 57.61 |
| MDS$_l$ | **97.38** | 72.32 | 74.75 | 68.91 | 68.98 | 55.41 | 83.29 | 75.24 | 63.41 | 38.92 | 77.56 | 62.16 |
| MDS$_{all}$ | 89.17 | 72.66 | **85.64** | 71.49 | 72.45 | 60.89 | **95.55** | 89.42 | 26.06 | 30.01 | 73.77 | 64.89 |
| REACT | 95.47 | 79.70 | 60.71 | 61.46 | 66.03 | 54.24 | 83.67 | 89.82 | 71.79 | 63.91 | 75.53 | 69.83 |
| GRADNORM | 90.85 | 75.53 | 65.17 | 56.52 | 72.19 | **65.57** | 85.00 | 89.39 | 69.59 | 54.45 | 76.56 | 68.29 |
| EBO | 95.52 | 73.8 | 59.72 | 59.59 | 65.91 | 57.72 | 83.83 | 91.14 | 71.27 | 61.55 | 75.25 | 68.76 |
| $D_\alpha$ | 91.27 | 67.95 | 58.62 | 61.44 | 64.95 | 55.65 | 81.57 | 87.43 | 72.49 | 67.15 | 73.78 | 67.92 |
| DICE | 55.7 | 74.45 | 78.29 | 58.76 | 77.84 | 59.43 | 86.67 | 91.38 | 61.23 | 59.97 | 71.95 | 68.8 |
| VIM | 95.76 | **81.55** | 56.85 | 62.91 | 61.01 | 53.26 | 79.79 | 87.00 | 68.45 | 49.01 | 72.37 | 66.75 |
| ASH | 95.52 | 73.89 | 59.72 | 69.61 | 65.91 | 57.94 | 83.83 | 91.13 | 71.27 | 61.65 | 75.25 | 70.84 |
| SHE | 90.98 | 76.71 | 72.15 | 68.37 | 67.19 | 63.87 | 82.38 | 89.79 | 60.92 | 60.91 | 74.72 | 71.93 |
| RELATION | 93.61 | 76.06 | 68.77 | 67.55 | 65.67 | 57.58 | 79.95 | 87.80 | 64.79 | 59.49 | 74.56 | 69.70 |
| ENS-V (ours) | 94.97 | 79.42 | 82.67 | 74.85 | **78.45** | 60.55 | 92.76 | 91.08 | 73.27 | 53.78 | **84.42** | 71.93 |
| ENS-R (ours) | 95.00 | 80.42 | 80.79 | **75.21** | 76.56 | 62.38 | 92.17 | 90.56 | **74.79** | 60.79 | 83.86 | **73.87** |
| ENS-F (ours) | 95.08 | 79.16 | 79.05 | 69.32 | 75.02 | 59.89 | 91.57 | **91.59** | 72.55 | 61.41 | 82.65 | 72.27 |

(OOD) detection. We specifically focus on *multiple labels* samples, which consist of at least two distinct classes from the training set occupying a substantial portion of the image.

Consider a classifier trained to differentiate dogs from cats; the label of an image featuring a dog next to a cat is ambiguous, and classifying it as either dog or cat is erroneous. In safety-critical applications, this issue could result in unpredictable outcomes and requires precautionary measures, such as human intervention. For example, a system tasked with identifying dangerous objects could misclassify an image featuring both a knife and a hat as safe by identifying the image as a hat.

The CoComageNet benchmark is constructed as a subset of the CoCo dataset [50], specifically, the 2017 training images. We identify 17 CoCo classes that have equivalent counterparts in ImageNet (please refer to appendix A for a comprehensive list of the selected CoCo classes and their ImageNet equivalents). We then filter the CoCo images to include only those containing at least two different classes among the selected 17. We calculate the total area occupied by each selected class and order the filtered images based on the portion of the image occupied by the second-largest class. The top 2000 images based on this metric constitute CoComageNet. By design, each image in CoComageNet contains at least two distinct ImageNet classes occupying substantial areas.

Although CoComageNet was developed to study the detection of multiple label images, it also exhibits other less easily characterized shifts, such as differences in the properties of ImageNet and CoCo images, and the fact that CoComageNet comprises only 17 of the 1000 ImageNet classes. To isolate the effect of multiple labels, we also construct CoComageNet-mono, a similar subset of CoCo that contains only one of the selected ImageNet classes (see appendix A for details).

As shown in appendix A, detection performances for all baselines on CoComageNet-mono are near random, demonstrating that detection of CoComageNet is primarily driven by the presence of multiple labels. Finally, to reduce the impact of considering only a subset of ImageNet classes, we evaluate detection methods using in-distribution ImageNet samples from the selected classes only.

## 3   Detection Methods

In this study, our focus is predominantly on methods that do not require training or fine-tuning using OOD samples. This consideration closely aligns with real-world applications where OOD samples are typically not known *a priori*. Additionally, the practice of fine-tuning or training on specific types of distribution shifts heightens the risk of overfitting them.

**Evaluated Methods:** We assess the broad OOD detection capabilities of a large number of methods including ASH [17], SHE [91], RELATION [40], REACT [74], VIM [82], GRADNORM [35], EBO [52], DICE [73], DOCTOR [24], CADET [25], ODIN [49], and Mahalanobis Distance (MDS) [46]. Fur-

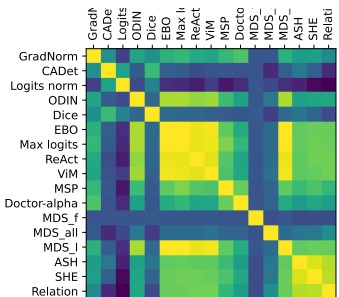 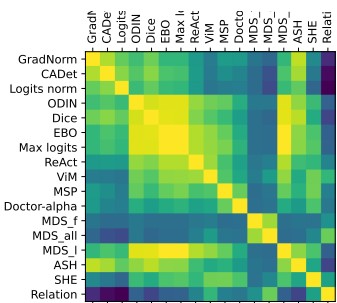

Figure 3: Covariance matrices of detection scores in-distribution for ViT (left) and ResNet-50 (right).

thermore, we explore three statistics widely applied in post-hoc OOD detection: maximum softmax probabilities (MSP), maximum of logits, and logit norm.

In the case of CADET, we solely utilize the intra-similarity score $m_{\text{in}}$ with five transformations to minimize computational demands. For DOCTOR, we employ $D_\alpha$ in the Totally Black Box (TBB) setting, disregarding $D_\beta$ as it is functionally equivalent to MSP in the TBB setting when rescaling the detection threshold is accounted for (resulting in identical AUC scores). ODIN typically depends on the fine-tuning of the perturbation budget $\epsilon$ and temperature $T$ on OOD samples. To bypass this requirement, we use default values of $\epsilon = 0.0014$ and $T = 1000$. These default parameters, having been tuned on a variety of datasets and models, have demonstrated robust generalization capabilities. Nevertheless, it should be noted that the choice of these values, despite being considered reasonable, does represent a caveat, as they were initially determined by tuning OOD detection of novel classes.

In its standard form, the Mahalanobis detection method computes the layer-wise Mahalanobis distance, followed by training a logistic regressor on OOD samples to facilitate detection based on a weighted average of these distances. To eliminate the need for training on OOD samples, we consider three statistics derived from Mahalanobis distances: the Mahalanobis distance on the output of the first layer block (MDS$_\text{f}$), the Mahalanobis distance on the output of the last layer (MDS$_\text{l}$), and the Mahalanobis distance on the output of all layers averaged with equal weights (MDS$_\text{all}$). For the Vision Transformer (ViT), we focus on MDS on the class token, disregarding patch tokens.

**Generative Modeling for Detection:** Consider $\mathcal{X}$ as a data distribution with a support set denoted as $X$, and let $h : X \to \mathbb{R}^d$ be a map that extracts features from a predetermined neural network. The function $h(x)$ can be defined arbitrarily; for instance, it could be the logits that the network computes on a transformation of a sample $x$, or the concatenation of outputs from different layers, among other possibilities. However, generative modeling in the input space (i.e., when $h$ is the identity function) is generally difficult due to the exceedingly high dimensionality and intricate structure of the data.

To avoid modeling the density in the high dimensional input space, we can use as proxy the density $p_{x \sim \mathcal{X}}(h(x))$ in the latent space $h(\mathcal{X})$. A generative model of $h$ is tasked with learning the distribution $p_{x \sim \mathcal{X}}(h(x))$, using a training set $(x_i)_{i \leq N}$ that comprises independently sampled instances from $\mathcal{X}$, and the log-density of the generative model can then directly be used as detection score.

A significant number of detection methods devise heuristic scores on $h$ with the aim of maximizing detection performances on specific benchmarks, while often arbitrarily discarding information that could potentially be beneficial for other distribution shifts. In contrast, generative models learn an estimator of the likelihood of $h(x)$ without discarding any information. Their detection performances are only constrained by the information extracted by $h$ and, naturally, their proficiency in learning its distribution. This inherent characteristic makes generative models particularly suitable for broad Out-of-Distribution (OOD) detection. By learning the comprehensive distribution of $h$, these models negate the bias associated with engineering detection scores against specific distribution shifts.

**Gaussian Mixtures Ensembling:** Gaussian Mixture Models (GMMs) are a versatile tool for learning a distribution of the form $x \sim \sum_i^n \pi_i \mathcal{N}(\mu_i, \Sigma_i)$, where $n$ is the number of components, $\pi$, $\mu$ and $\Sigma$ are the parameters of the GMM and are learned with the Expectation-Maximization (EM) algorithm.

GMM-based generative modeling of neural network behaviors to facilitate detection has been previously reported [9]. Methods that are based on the Mahalanobis distance bear similarity to this approach insofar as the layer-wise Mahalanobis score can be interpreted as the likelihood of the layer output for class-dependent Gaussian distributions, which are learned from the training set.

Despite these advantages, such methods encounter the formidable challenge of learning generative models of the network's high dimensional representation space, a task made more difficult due to the curse of dimensionality. In response to this challenge, we propose the learning of a Gaussian mixture of the scores computed by existing OOD detection methods. While this approach still relies on heuristic scores, it presents an ensemble method that is able to amalgamate their respective information, while maintaining the dimension of its underlying variables at a significantly low level. As a result, it achieves a favorable tradeoff between the generative modeling of high dimensional feature spaces and the heuristic construction of one-dimensional detection scores.

In addition to integrating their detection capabilities, this approach is adept at identifying atypical realizations of the underlying scores, even in situations where the marginal likelihood of each score is high, but their joint likelihood is low.

To make our method as general as possible, **we do not assume access to OOD samples to select which scores to use** as variables of our GMM. We present in Figure 3 the covariance matrices of the different scores on a held-out validation set of ImageNet. To minimize redundancy, we avoid picking multiple scores that are highly correlated on clean validation data. To decide between highly correlated scores, we opt for the ones with highest in-distribution error detection performance (see first two columns of Table 8). Moreover, we discard logit norm and $MDS_f$ due to their near-random error detection performance in-distribution. Given that score correlation varies between ViTs and ResNets, as evidenced in Figure 3, we derive two distinct sets of scores. We also propose a computationally efficient alternative based on methods with minimal overhead costs:

**Ens-V** (ViT) = {GRADNORM, ODIN, $MDS_{all}$, $MDS_l$, CADET, DICE, MSP, MAX LOGITS},

**Ens-R** (ResNet) = {GRADNORM, ODIN, $MDS_{all}$, $MDS_l$, CADET, REACT, VIM, $D_\alpha$},

**Ens-F** (Fast) = {MSP, MAX LOGITS, $MDS_{all}$, $MDS_l$, EBO}.

We train the GMM on the correctly-classified samples of a held-out validation set of 45,000 samples. This is essential as misclassified samples may produce atypical values of the underlying scores despite being in-distribution, which is demonstrated by the high in-distribution error detection AUC of selected scores. Finally, we train the GMM for a number of components $n \in \{1, 2, 5, 10, 20\}$ and select $n = 10$ which maximizes the in-distribution error detection performances (see appendix C).

## 4 Evaluation

We assess performance using the widely accepted area under the curve (AUC) metric for two distinct pretrained models: ResNet-50 (RN50) and Vision Transformer (ViT). While it is standard to also report other metrics such as FPR@95 and AUPR, we find these metrics to be redundant with AUC and omit them for clarity. All evaluations are conducted on a single A100 GPU, with the inference time normalized by the cost of a forward pass (cf. App. B).

Our empirical results in the Distribution Shift Detection (DSD) setting, which aims to detect any OOD sample, are presented in Table 2. Results for the error detection setting, where the objective is to detect misclassified OOD samples against correctly classified in-distribution samples, are exhibited in Appendix D (Table 8). The results for each distribution shift type are averaged over the corresponding benchmark. Detailed performances and model accuracy for each dataset are offered in Appendix D (where applicable). In the error detection setting, we conduct evaluations against adversarial attacks, corruptions, and in-distribution. The latter pertains to predicting classification errors on authentic ImageNet inputs. Please note that error detection is inapplicable to novel classes and multi-labels where correct classifications are undefined, and we do not consider error detection on synthetic images as it lacks clear motivation.

**Existing methods:** A striking observation is the inconsistency of recent detection methods in the broad OOD setting. Methods that excel on adversarial attacks tend to underperform on multi-label detection, and vice versa. Each of the baselines exhibits subpar performance on at least one

distribution shift, and almost all of them are Pareto-optimal. This underscores the necessity for broader OOD detection evaluations to inform the design of future methods.

We observe that while detection performances are generally superior when utilizing a ViT backbone, a finding consistent with previous studies [82], the difference is method-dependent. For instance, $MDS_l$ ranks as the best baseline on ViT (when averaged over distribution shift types), but it is the third-worst with a ResNet-50.

We further observe that many methods significantly outperform a random choice in the detection of synthetic images, regardless of the generation methods used (see Appendix D). This suggests that despite recent advancements in generative models, the task remains feasible.

Interestingly, the performance of various methods relative to others is remarkably consistent between the DSD and error detection settings, applicable to both adversarial attacks and corruptions. This consistency implies a strong correlation between efficiently detecting OOD samples and detecting errors induced by distribution shifts, suggesting that there may not be a need to compromise one objective for the other.

**Ensemble:** Our ensemble method surpasses all baselines when averaged over distribution shift types, evident in both the DSD and error detection settings, and consistent across both ViT and ResNet-50 backbones. With the exception of error detection with ResNet-50, where Doctor-alpha yields comparable results, our ensemble method consistently demonstrates significant improvements over the best-performing baselines. Specifically, in the DSD setting, ENS-V and ENS-R secure improvements of 6.86% and 4.04% for ViT and ResNet-50, respectively.

While the ensemble detection rarely surpasses the best baselines for a specific distribution shift type, it delivers more consistent performances across types, which accounts for its superior averaged AUC. This finding endorses the viability of our approach for broad OOD detection.

Despite the notable computational overhead for ENS-V and ENS-R (up to $13.92\times$ the cost of a forward pass for ENS-V with ResNet-50, as detailed in Appendix B), the inference of ENS-F atop a forward pass only adds a modest 19% to 25% overhead, thus striking a reasonable balance between cost and performance.

Interestingly, ENS-F trails only slightly in terms of performance in the DSD setting. In the error detection setting, ENS-F unexpectedly delivers the best results for both ViT and ResNet.

# 5   Related Work

In this work, we study the detection of out-of-distribution (OOD) samples with a broad definition of OOD, encompassing various types of distribution shifts. Our work intersects with the literature in OOD detection, adversarial detection, and synthetic image detection. We also provide a brief overview of uncertainty quantification methods that can be leveraged to detect errors induced by distribution shifts.

**Label-based OOD detection** has been extensively studied in recent years under different settings: anomaly detection [8, 66, 71], novelty detection [60, 67], open set recognition [58, 22, 6], and outlier detection [80, 32, 3]. Most existing methods can be categorized as either density-based [47, 38], reconstruction-based [16, 86], classification-based [81, 79] or distance-based [89, 76]. Methods can further be divided based on whether they require pre-processing of the input, specific training schemes, external data or can be used a post-processors on any trained model. See Yang et al. [84] for a complete survey.

**Adversarial detection** is the task of detecting adversarially perturbed inputs. Most existing methods require access to adversarial samples [2, 93, 54, 61, 55, 4, 59], with some exceptions [33, 5, 25]. Since adversarial training does not transfer well across attacks [37], adversarial detection methods that assume access to adversarial samples are also unlikely to generalize well. Unfortunately, Carlini and Wagner [10] have shown that recent detection methods can be defeated by adapting the attack's loss function. Thus, attacks targeted against the detector typically remain undetected. However, adversarial attacks transfer remarkably well across models [11, 23], which makes deployed systems vulnerable even when the attacker does not have access to the underlying model. Detectors thus make systems more robust by requiring targeted attack designs.

**Synthetic image detection** is the detection of images that have been artificially generated. Following the rapid increase in generative models' performances and popularity [68, 64, 70], many works have addressed the task of discriminating synthetic images from genuine ones [51]. They are generally divided between image artifact detection [51, 14, 92] and data-drive approaches [83]. Since generative models aim at learning the genuine distribution, their shortcomings only permit detection. As generative models improve, synthetic images may become indistinguishable from genuine ones.

**Uncertainty quantification** (UQ) for deep learning aims to improve the estimation of neural network confidences. Neural networks tend to be overconfident even on samples far from the training distribution [63]. By better estimating the confidence in the network's predictions, uncertainty quantification can help detect errors induced by distribution shifts. See Abdar et al. [1], Kabir et al. [39], Ning and You [65] for overviews of UQ in deep learning.

**Detection of multiple types of distribution shifts** has been addressed by relatively few prior works. The closest work in the literature is probably Guille-Escuret et al. [25] and Lee et al. [46] which aims at simultaneously detecting novel classes and adversarial samples. In comparison, this work evaluates detection methods on five different types of distribution shifts. To the best of our knowledge, it is the first time that such broad OOD detection is studied in the literature.

# 6   Conclusion

We have evaluated recent OOD detection methods on BROAD, a novel diversified benchmark we introduced spanning 5 different distribution shift types, and found their performances unreliable. Due to the literature focusing on specific distribution shifts, existing methods often fail to detect samples of certain out-of-distribution shifts.

We encourage future work to consider more varied types of OOD samples for their detection evaluations, so that future methods will not see their success limited to unexpected inputs that are expected. Moreover, while setting ImageNet as an in-distribution can yield insights on a large class of models and applications, future work should consider additional in-distributions to expand BROAD's coverage, including different modalities such as text and audio.

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

# A CoComageNet

Table 3: CoCo and ImageNet classes used for CoComageNet and CoComageNet-mono.

| CoCo | | ImageNet | |
| ID | Name | ID | Name |
| --- | --- | --- | --- |
| 24 | Zebra | n02391049 | Zebra |
| 27 | Backpack | n02769748 | Backpack, back pack, knapsack, packsack, rucksack, haversack |
| 28 | Umbrella | n04507155 | Umbrella |
| 35 | Skis | n04228054 | Ski |
| 38 | Kite | n01608432 | Kite |
| 47 | Cup | n07930864 | Cup |
| 52 | Banana | n07753592 | Banana |
| 55 | Orange | n07747607 | Orange |
| 56 | Broccoli | n07714990 | Broccoli |
| 59 | Pizza | n07873807 | Pizza, pizza pie |
| 73 | Laptop | n03642806 | Laptop, laptop computer |
| 74 | Mouse | n03793489 | Mouse, computer mouse |
| 75 | Remote | n04074963 | Remote control, remote |
| 78 | Microwave | n03761084 | Microwave, microwave oven |
| 80 | Toaster | n04442312 | Toaster |
| 82 | Refrigerator | n04070727 | Refrigerator, icebox |
| 86 | Vase | n04522168 | Vase |

We here provide additional information related to the CoComageNet and CoComageNet-mono datasets, together referred to as CoComageNet.

Table 3 lists the classes used for CoComageNet with their corresponding IDs and names for both CoCo and ImageNet. These classes were automatically selected by finding matches between CoCo names and ImageNet IDs understood as WordNet synsets [20]. Only exact matches were considered; hyponyms and hypernyms were excluded. While one could argue for more classes to be added to this list, we believe that those present on this list are "safe".

CoComageNet, introduced in section 2.5, aims to induce a distribution shift due to the presence of multiple classes. However, it is also affected by the distributional variations between ImageNet and CoCo, such as different angles, distances, brightness, etc.

To alleviate this issue, we introduce the sister dataset CoComageNet-mono by selecting 2000 different images from the same CoCo 2017 training dataset. Disregarding any "Person" CoCo label, we only keep the images whose labels belong to a single CoCo class, and only if that class is one of the 17 listed in table 3. For example, the photograph of a person holding several bananas satisfies these conditions (disregarding the person, the labels are all in the same "banana" class) while one with a cat next to a banana does not (even though "cat" is not listed in table 3, it is a CoCo class). For classes with less than 157 images left, we add all these images to CoComageNet-mono. For the other classes, we

Table 4: Detection AUC of ResNet-50 and ViT for different detection scores against CoComageNet and CoComageNet-mono

| | CoComageNet | | CoComageNet-mono | |
| | ViT | RN-50 | ViT | RN-50 |
| --- | --- | --- | --- | --- |
| CADet $m_{in}$ | 54.24 | 56.88 | 52.24 | 50.8 |
| ODIN | 70.75 | 64.46 | 55.27 | 53.72 |
| Max logits | 71.63 | 62.79 | 56.15 | 53.32 |
| Logit norm | 36.32 | 48.05 | 51.03 | 55.26 |
| MSP | 71.93 | 67.52 | 53.16 | 53.32 |
| $MDS_f$ | 63.43 | 36.81 | 61.38 | 48.45 |
| $MDS_l$ | 63.41 | 38.92 | 58.32 | 50.97 |
| $MDS_{all}$ | 26.06 | 30.01 | 46.66 | 47.06 |
| ReAct | 71.79 | 63.91 | 57.81 | 58.73 |
| GradNorm | 69.59 | 54.45 | 56.34 | 53.43 |
| EBO | 71.27 | 61.55 | 56.65 | 53.31 |
| $D_\alpha$ | 72.49 | 67.15 | 53.41 | 53.08 |
| Dice | 61.23 | 59.97 | 57.35 | 53.00 |
| ViM | 68.45 | 49.01 | 57.06 | 54.54 |
| ASH | 71.27 | 61.65 | 56.65 | 53.32 |
| SHE | 60.92 | 60.91 | 53.55 | 53.77 |
| Relation | 64.79 | 59.49 | 52.26 | 49.91 |
| Ens-V (us) | 73.22 | 61.29 | 59.21 | 59.06 |
| Ens-R (us) | 74.79 | 61.01 | 60.29 | 56.75 |
| Ens-F (us) | 72.65 | 61.42 | 58.92 | 58.34 |

Table 5: Normalized inference time.

|  | ViT | RN-50 |
| --- | --- | --- |
| Forward | 1.00 | 1.00 |
| Cadet $m_{\mathrm{in}}$ | 5.04 | 5.15 |
| Odin | 3.22 | 2.94 |
| max logit | 1.01 | 1.00 |
| logit norm | 1.01 | 1.00 |
| MSP | 1.01 | 1.00 |
| $MDS_f$ | 1.23 | 1.17 |
| $MDS_l$ | 1.23 | 1.17 |
| $MDS_{all}$ | 1.23 | 1.17 |
| ReAct | 1.11 | 1.06 |
| GradNorm | 2.28 | 3.86 |
| EBO | 1.01 | 1.03 |
| $D_\alpha$ | 1.01 | 1.06 |
| Dice | 1.03 | 1.09 |
| ViM | 3.64 | 2.03 |
| ASH | 1.03 | 1.05 |
| SHE | 1.03 | 1.00 |
| Relation | 1.01 | 1.03 |
| Ens-V (us) | 11.53 | 13.92 |
| Ens-R (us) | 10.25 | 10.92 |
| Ens-F (us) | 1.25 | 1.19 |

sort the images of each class according to the proportion of the image taken by that class, and add to CoComageNet-mono the top 157 by that metric (top 158 for the two most populated classes), for a total of 2000 images.

Table 4 shows the detection performances of all baselines and our method against CoComageNet and CoComageNet-mono. Detection performances on CoComageNet-mono are generally close to 50% (corresponding to random guess) which shows that the distribution shift between ImageNet and CoCo has limited influence on the detection scores of our baselines. In comparison, detection scores are generally significantly further from 50% on CoComageNet, showing it is indeed the presence of multiple classes that drives detection in the case of CoComageNet.

## B  Computation time

Table 5 presents the computation time of each method, normalized by the cost of a forward pass. Note that when normal inference is needed to compute the score, its computation time is included in the inference time. Therefore, running Ens-S on top of classification only has an additional overhead of 25% for ViT and 19% for ResNet.

## C  Number of components

We present in Table 6 and Table 7 the in-distribution error detection AUCs that were used to pick the number of components $n$ of the Gaussian mixture. We observe that the number of components has a low impact on performances, and that in-distribution error detection AUC has a clear correlation with broad OOD detection performances, making it an adequate metric to determine the number of components.

## D  Complete results

In this section, we present the error detection AUC in Table 8. Moreover, we provide in Table 9 to Table 15 the detection AUC of all methods against each dataset separately, both in the DSD and the error detection setting.

Table 6: In-distribution error detection AUC and OOD detection AUC averaged over distribution shift types, for a ResNet-50 using Ensemble-ResNet and using n Gaussian components.

| n | In-dist error detection | Avg OOD detection |
|---|---|---|
| 1 | 74.99 | 71.85 |
| 2 | 76.02 | 73.08 |
| 5 | 77.13 | **73.51** |
| 10 | **77.24** | 73.46 |
| 20 | 75.16 | 71.44 |

Table 7: In-distribution error detection AUC and OOD detection AUC averaged over distribution shift types, for a ViT using Ensemble-ViT and using n Gaussian components.

| n | In-dist error detection | Avg OOD detection |
|---|---|---|
| 1 | 82.41 | 82.91 |
| 2 | 82.34 | 83.20 |
| 5 | 82.59 | 83.61 |
| 10 | **82.61** | **83.66** |
| 20 | 82.19 | 81.80 |

Table 8: Error detection AUC for Visual Transformer and ResNet-50.

| | In-distribution | | Adv. Attacks | | Corruptions | | Average | |
|---|---|---|---|---|---|---|---|---|
| | ViT | RN50 | ViT | RN50 | ViT | RN50 | ViT | RN50 |
| CADET $m_{in}$ | 54.63 | 56.50 | 70.02 | 67.85 | 81.55 | 90.11 | 68.73 | 71.49 |
| ODIN | 75.02 | 75.79 | 56.98 | 62.96 | 90.83 | 95.02 | 74.28 | 77.92 |
| MAX LOGIT | 80.64 | 77.55 | 67.67 | 68.37 | 95.55 | **96.84** | 81.29 | 80.92 |
| LOGIT NORM | 36.83 | 50.11 | 34.05 | 55.94 | 33.65 | 84.37 | 34.84 | 63.47 |
| MSP | **89.16** | 86.31 | 70.29 | 74.04 | 95.75 | 95.93 | 85.07 | 85.43 |
| MDS$_f$ | 48.23 | 51.25 | 68.18 | 56.66 | 30.70 | 76.47 | 49.04 | 61.46 |
| MDS$_l$ | 74.92 | 55.53 | 82.98 | 72.39 | 96.39 | 75.85 | 84.76 | 67.92 |
| MDS$_{all}$ | 54.93 | 54.69 | 89.44 | 74.65 | **99.1** | 89.90 | 81.16 | 73.08 |
| REACT | 77.18 | 73.28 | 68.00 | 69.32 | 94.81 | 95.01 | 80.00 | 79.20 |
| GRADNORM | 68.01 | 58.07 | 70.92 | 62.49 | 94.00 | 92.73 | 77.64 | 71.10 |
| EBO | 78.35 | 76.02 | 66.63 | 67.63 | 95.01 | 96.61 | 80.00 | 80.09 |
| $D_\alpha$ | 89.00 | **86.50** | 70.37 | 73.97 | 95.96 | 96.26 | 85.11 | 85.58 |
| DICE | 56.91 | 70.00 | 80.6 | 65.71 | 89.13 | 95.53 | 75.55 | 77.08 |
| VIM | 75.72 | 73.74 | 63.37 | 69.83 | 91.64 | 93.03 | 76.91 | 78.87 |
| ASH | 78.35 | 77.42 | 66.63 | 78.06 | 95.01 | 96.74 | 80.00 | 82.94 |
| SHE | 78.96 | 67.03 | 83.54 | 75.02 | 95.7 | 94.20 | 86.07 | 78.75 |
| RELATION | 76.91 | 74.21 | 79.1 | 76.28 | 94.18 | 95.00 | 83.40 | 81.83 |
| ENS-V (ours) | 82.61 | 72.81 | **89.52** | 80.47 | 98.27 | 94.73 | 90.13 | 82.67 |
| ENS-R (ours) | 83.69 | 77.24 | 88.17 | **83.79** | 97.84 | 95.92 | 89.90 | 85.65 |
| ENS-F (ours) | 85.84 | 79.20 | 88.72 | 82.60 | 98.41 | 96.49 | **90.99** | **86.10** |

Table 9: AUC for OOD detection in DSD setting for ResNet on novel classes datasets.

| | iNat | OI-O | INet-O |
|---|---|---|---|
| CADet $m_{\text{in}}$ | 88.08 | 74.41 | 37.88 |
| ODIN | 91.19 | 88.26 | 41.28 |
| Max logits | 91.17 | 89.14 | 40.69 |
| Logits norm | 55.98 | 66.19 | 35.68 |
| MSP | 88.34 | 84.85 | 28.55 |
| MDS$_f$ | 63.14 | 61.70 | 65.71 |
| MDS$_l$ | 63.18 | 69.32 | **84.45** |
| MDS$_{\text{all}}$ | 61.42 | 72.81 | 83.74 |
| ReAct | **96.39** | **90.33** | 52.37 |
| GradNorm | 93.90 | 84.79 | 47.9 |
| EBO | 90.63 | 89.03 | 41.75 |
| $D_\alpha$ | 89.43 | 85.84 | 28.57 |
| Dice | 92.50 | 88.25 | 42.61 |
| ViM | 88.15 | 88.05 | 68.45 |
| ASH | 90.53 | 88.95 | 42.19 |
| SHE | 91.69 | 86.11 | 52.34 |
| Relation | 87.09 | 84.39 | 56.69 |
| Ens-V (us) | 85.50 | 81.96 | 70.81 |
| Ens-R (us) | 89.40 | 86.11 | 65.74 |
| Ens-F (us) | 88.06 | 86.64 | 62.79 |

Table 10: AUC for OOD detection in DSD setting for ResNet on synthetic datasets.

| | Biggan | diffusion |
|---|---|---|
| Accuracy % | 88.61 | 47.38 |
| CADet $m_{\text{in}}$ | 63.18 | 48.12 |
| ODIN | 44.46 | 78.51 |
| Max logits | 42.14 | 73.15 |
| Logits norm | 59.73 | 59.21 |
| MSP | 41.37 | 69.81 |
| MDS$_f$ | 38.25 | 74.11 |
| MDS$_l$ | 38.95 | 71.86 |
| MDS$_{\text{all}}$ | 40.65 | **81.12** |
| ReAct | 34.83 | 73.64 |
| GradNorm | **75.65** | 55.49 |
| EBO | 42.35 | 73.08 |
| $D_\alpha$ | 41.11 | 70.18 |
| Dice | 51.23 | 67.63 |
| ViM | 32.46 | 74.06 |
| ASH | 42.69 | 73.18 |
| SHE | 63.74 | 64.00 |
| Relation | 47.29 | 67.87 |
| Ens-V (us) | 46.19 | 74.91 |
| Ens-R (us) | 47.73 | 77.02 |
| Ens-F (us) | 41.75 | 78.03 |

Table 11: AUC for OOD detection in DSD setting for ResNet on corruptions datasets.

| | defocus blur | Gaussian noise | snow | brightness |
|---|---|---|---|---|
| Accuracy % | 15.04 | 5.68 | 15.58 | 55.64 |
| CADet $m_{\text{in}}$ | 96.17 | 95.24 | 86.47 | 70.70 |
| ODIN | 97.17 | 99.01 | 89.22 | 68.66 |
| Max logits | 96.54 | 97.65 | 93.76 | 75.53 |
| Logits norm | 87.48 | 90.37 | 86.04 | 67.36 |
| MSP | 94.05 | 94.88 | 87.57 | 70.32 |
| $\text{MDS}_f$ | 46.35 | 98.34 | 88.67 | 72.70 |
| $\text{MDS}_l$ | 68.85 | 96.44 | 78.72 | 56.96 |
| $\text{MDS}_{\text{all}}$ | 92.87 | 99.52 | 91.28 | 73.99 |
| ReAct | 94.88 | 97.01 | 94.09 | 73.31 |
| GradNorm | 98.02 | 96.97 | 90.06 | 72.51 |
| EBO | 96.62 | 97.86 | 94.29 | 75.78 |
| $D_\alpha$ | 94.66 | 95.66 | 88.61 | 70.80 |
| Dice | 97.81 | 98.08 | 93.47 | **76.16** |
| ViM | 83.9 | 97.11 | 94.19 | 72.80 |
| ASH | 96.58 | 97.82 | 94.24 | 75.86 |
| SHE | 97.63 | 97.19 | 90.81 | 73.53 |
| Relation | 95.08 | 96.39 | 90.64 | 69.10 |
| Ens-V (us) | 97.39 | **99.68** | 94.62 | 72.64 |
| Ens-R (us) | 97.04 | 99.57 | 93.58 | 72.04 |
| Ens-F (us) | **98.13** | 99.41 | **94.96** | 73.84 |

Table 12: AUC for OOD detection in DSD setting for ResNet on adversarial attacks dataset. PGD ResNet denotes PGD computed against ResNet (hence white box), and PGD ViT against a separate ViT model (hence black box).

| | PGD ResNet | AA ResNet | PGD ViT | AA ViT |
|---|---|---|---|---|
| Accuracy % | 2.2 | 25.8 | 68.12 | 43.2 |
| CADet $m_{\text{in}}$ | 45.37 | 71.11 | **64.86** | 68.25 |
| ODIN | 12.91 | 79.70 | 54.98 | 70.18 |
| Max logits | 18.54 | **84.50** | 59.42 | **76.01** |
| Logits norm | 13.47 | 70.21 | 58.31 | 65.30 |
| MSP | 30.82 | 82.23 | 58.02 | 73.59 |
| $\text{MDS}_f$ | 71.17 | 55.59 | 44.73 | 48.67 |
| $\text{MDS}_l$ | 88.19 | 74.36 | 46.81 | 66.26 |
| $\text{MDS}_{\text{all}}$ | 86.00 | 81.05 | 46.83 | 72.07 |
| ReAct | 33.02 | 82.62 | 55.62 | 74.56 |
| GradNorm | 15.62 | 77.67 | 63.52 | 69.25 |
| EBO | 18.52 | 84.46 | 59.42 | 75.97 |
| $D_\alpha$ | 30.62 | 82.90 | 58.13 | 74.11 |
| Dice | 16.55 | 82.78 | 61.34 | 74.35 |
| Vim | 39.40 | 82.85 | 54.30 | 75.10 |
| ASH | 57.67 | 84.71 | 59.89 | 76.18 |
| SHE | 56.06 | 82.25 | 62.41 | 72.77 |
| Relation | 59.25 | 82.50 | 55.41 | 73.04 |
| Ens-V (us) | **91.56** | 81.12 | 54.91 | 71.81 |
| Ens-R (us) | 89.19 | 82.88 | 55.28 | 73.48 |
| Ens-F (us) | 66.86 | 83.38 | 54.13 | 72.89 |

Table 13: AUC in error detection setting for ResNet.

| | In-Dist | Adv. Attacks | | | | Corruptions | | | |
|---|---|---|---|---|---|---|---|---|---|
| | | PGD RN | AA RN | PGD ViT | AA ViT | blur | noise | snow | bright. |
| CADet $m_{in}$ | 56.50 | 46.09 | 79.56 | 68.22 | 77.52 | 97.3 | 96.16 | 89.08 | 77.91 |
| ODIN | 75.79 | 15.23 | 90.02 | 60.53 | 86.05 | 99.05 | 99.68 | 94.42 | 86.91 |
| Max logits | 77.55 | 21.99 | **94.58** | 65.26 | 91.63 | 98.9 | 99.10 | 97.66 | 91.68 |
| Logits norm | 50.11 | 13.94 | 76.73 | 59.98 | 73.11 | 88.88 | 90.96 | 87.52 | 70.13 |
| MSP | **86.31** | 38.06 | 94.06 | 71.87 | 92.16 | 98.34 | 97.97 | 94.65 | 92.76 |
| MDS$_f$ | 51.25 | 71.44 | 54.82 | 45.66 | 54.72 | 46.12 | 98.44 | 88.89 | 72.44 |
| MDS$_l$ | 55.53 | **88.45** | 80.97 | 46.25 | 73.87 | 68.74 | 96.35 | 79.15 | 59.14 |
| MDS$_{all}$ | 54.69 | 85.67 | 85.00 | 47.96 | 79.96 | 93.14 | 99.54 | 91.67 | 75.23 |
| ReAct | 73.28 | 37.80 | 92.24 | 58.27 | 88.95 | 97.25 | 98.28 | 97.05 | 87.46 |
| GradNorm | 58.07 | 16.41 | 86.52 | 67.03 | 79.99 | 98.85 | 97.84 | 93.01 | 81.21 |
| EBO | 76.02 | 21.82 | 94.12 | 63.70 | 90.86 | 98.76 | 99.08 | **97.68** | 90.93 |
| $D_\alpha$ | 86.20 | 37.90 | 94.42 | 71.05 | **92.52** | 98.56 | 98.3 | 95.16 | **93.00** |
| Dice | 70.00 | 19.15 | 91.50 | 65.13 | 87.05 | 98.94 | 98.84 | 96.19 | 88.13 |
| ViM | 73.74 | 43.99 | 92.62 | 54.47 | 88.23 | 89.90 | 98.52 | 97.13 | 86.58 |
| ASH | 77.42 | 63.02 | 94.51 | 63.66 | 91.04 | 98.82 | 99.12 | 97.78 | 91.25 |
| SHE | 67.03 | 58.92 | 91.11 | 65.54 | 84.50 | 98.85 | 98.27 | 94.39 | 85.28 |
| Relation | 74.21 | 63.96 | 93.58 | 59.02 | 88.57 | 98.33 | 98.39 | 96.11 | 87.18 |
| Ens-V(us) | 72.81 | 80.72 | 86.67 | 72.03 | 82.44 | 98.40 | **99.80** | 96.59 | 84.12 |
| Ens-R (us) | 77.24 | 78.62 | 90.66 | 78.41 | 87.48 | 99.26 | 99.68 | 97.20 | 87.52 |
| Ens-F (us) | 79.20 | 73.25 | 90.45 | **79.09** | 87.61 | **99.28** | 99.73 | **97.68** | 89.26 |

Table 14: AUC for OOD detection in DSD setting for ViT.

| | iNat | OI-O | INet-O | PGD-R | AA-R | PGD-V | AA-V | Biggan | diff | blur | noise | snow | bright |
|---|---|---|---|---|---|---|---|---|---|---|---|---|---|
| Acc % | - | - | - | 77.00 | 65.22 | 0.46 | 50.66 | 86.28 | 55.77 | 42.09 | 42.85 | 56.82 | 76.12 |
| CADet | 8.30 | 24.83 | 29.61 | 63.06 | 77.29 | 60.64 | 67.48 | 68.72 | 50.92 | 98.12 | 72.45 | 78.37 | 69.72 |
| ODIN | 97.05 | 93.85 | 84.28 | 57.76 | 72.44 | 11.92 | 67.02 | 49.19 | 76.28 | 87.23 | 93.96 | 76.72 | 60.81 |
| Max logits | 98.65 | 97.06 | 90.04 | 63.68 | 76.44 | 24.82 | 73.96 | 54.87 | 77.28 | 94.24 | 90.97 | 83.20 | 66.00 |
| logits norm | 50.84 | 51.70 | 53.24 | 41.58 | 39.26 | 31.18 | 37.52 | 42.61 | 33.89 | 41.62 | 41.91 | 35.54 | 40.89 |
| MSP | 96.39 | 92.99 | 82.31 | 61.61 | 71.99 | 26.64 | 73.57 | 54.81 | 74.74 | 88.83 | 85.71 | 77.11 | 62.82 |
| MDS$_f$ | 66.73 | 53.68 | 39.63 | 68.59 | 77.31 | 68.76 | 56.24 | 49.04 | 60.79 | 40.95 | 62.54 | 07.00 | 15.39 |
| MDS$_l$ | **99.63** | **98.22** | **94.28** | 68.59 | 76.34 | **78.54** | 75.54 | 53.61 | 84.35 | 82.32 | 97.22 | 85.07 | 68.56 |
| MDS$_{all}$ | 90.37 | 89.88 | 87.25 | 80.06 | 91.74 | 78.49 | **92.26** | 54.09 | **90.81** | 99.74 | 99.99 | **96.14** | **86.31** |
| ReAct | 98.67 | 97.12 | 90.62 | 64.09 | 75.22 | 29.99 | 73.52 | 54.95 | 77.10 | 93.62 | 90.95 | 83.23 | 66.87 |
| GradNorm | 97.35 | 94.53 | 80.68 | 67.47 | 84.36 | 35.00 | 73.83 | 73.79 | 70.59 | 99.00 | 88.32 | 83.17 | 69.50 |
| EBO | 98.69 | 97.26 | 90.61 | 63.68 | 76.5 | 25.20 | 73.48 | 54.8 | 77.02 | 94.54 | 91.20 | 83.48 | 66.11 |
| $D_\alpha$ | 97.03 | 93.76 | 83.02 | 61.74 | 72.30 | 26.67 | 73.77 | 54.81 | 75.08 | 89.41 | 96.29 | 77.58 | 62.99 |
| Dice | 51.43 | 63.67 | 51.99 | **83.91** | 89.67 | 62.91 | 76.68 | **86.29** | 69.39 | 96.56 | 80.85 | 87.12 | 82.13 |
| ViM | 98.88 | 97.07 | 91.33 | 61.36 | 69.30 | 26.72 | 70.00 | 46.87 | 75.15 | 82.01 | 91.90 | 81.50 | 63.75 |
| ASH | 98.69 | 97.26 | 90.61 | 63.68 | 76.51 | 25.20 | 73.48 | 54.80 | 77.02 | 94.54 | 91.20 | 83.48 | 66.11 |
| SHE | 91.54 | 92.55 | 88.84 | 64.31 | 77.68 | 71.49 | 75.12 | 57.32 | 77.05 | 92.74 | 89.08 | 82.07 | 65.63 |
| Relation | 96.95 | 95.27 | 88.62 | 62.43 | 73.83 | 65.76 | 73.06 | 55.35 | 75.99 | 89.27 | 87.83 | 79.32 | 63.38 |
| Ens-V(us) | 99.00 | 96.28 | 89.64 | 75.48 | **92.23** | 73.67 | 89.28 | 71.27 | 85.62 | **99.89** | **99.99** | 92.11 | 79.06 |
| Ens-R (us) | 98.90 | 96.59 | 89.50 | 71.47 | 90.74 | 72.37 | 88.58 | 68.38 | 84.74 | 99.86 | **99.99** | 91.40 | 77.41 |
| Ens-F (us) | 98.42 | 96.55 | 90.27 | 73.46 | 90.33 | 64.72 | 87.70 | 64.37 | 85.66 | 99.33 | 99.97 | 90.87 | 76.11 |

Table 15: AUC in error detection setting for ViT.

| | In-Dist | Adv. Attacks | | | | Corruptions | | | |
|---|---|---|---|---|---|---|---|---|---|
| | | PGD-R | AA-R | PGD-V | AA-V | blur | noise | snow | brightness |
| CADet $m_{\text{in}}$ | 49.73 | 63.59 | 83.42 | 60.59 | 72.46 | 98.75 | 75.98 | 80.88 | 70.60 |
| ODIN | 75.02 | 60.76 | 85.12 | 13.26 | 68.76 | 95.20 | 98.41 | 89.54 | 80.18 |
| Max logits | 80.64 | 71.08 | 91.30 | 27.84 | 80.47 | 99.03 | 98.07 | 95.95 | 89.13 |
| Logits norm | 36.83 | 41.91 | 34.12 | 29.06 | 31.12 | 39.59 | 38.48 | 27.87 | 28.65 |
| MSP | **89.16** | 77.15 | 92.01 | 30.80 | 81.19 | 98.29 | 97.28 | 95.46 | 91.95 |
| MDS$_{\text{f}}$ | 48.23 | 67.86 | 81.36 | 68.65 | 54.84 | 40.29 | 61.60 | 06.84 | 14.08 |
| MDS$_{\text{l}}$ | 74.92 | 75.46 | 88.62 | **82.28** | 85.57 | 91.23 | 99.45 | 95.25 | 99.64 |
| MDS$_{\text{all}}$ | 54.93 | 83.08 | 95.24 | 81.01 | **98.42** | 99.80 | 99.99 | 96.92 | **99.68** |
| ReAct | 77.18 | 69.62 | 89.09 | 33.15 | 80.12 | 98.56 | 97.56 | 95.26 | 87.87 |
| GradNorm | 68.01 | 72.49 | 93.74 | 37.21 | 80.24 | 99.79 | 96.48 | 94.31 | 85.42 |
| EBO | 78.35 | 68.77 | 90.13 | 28.07 | 79.54 | 98.90 | 97.82 | 95.48 | 87.82 |
| $D_\alpha$ | 89.00 | 76.97 | 92.31 | 30.82 | 81.37 | 98.49 | 97.54 | 95.74 | 92.05 |
| Dice | 50.09 | 85.18 | 93.85 | 63.02 | 80.35 | 97.61 | 85.60 | 89.82 | 83.47 |
| ViM | 75.72 | 65.66 | 82.18 | 29.50 | 76.14 | 90.59 | 97.81 | 93.58 | 84.59 |
| ASH | 78.35 | 68.77 | 90.13 | 28.06 | 79.54 | 98.90 | 97.82 | 95.48 | 87.82 |
| SHE | 78.96 | 76.44 | 93.13 | 76.35 | 88.22 | 98.88 | 97.64 | 96.19 | 90.09 |
| Relation | 76.91 | 71.08 | 90.38 | 70.55 | 84.39 | 97.97 | 97.21 | 94.58 | 86.96 |
| Ens-V(us) | 82.61 | 92.73 | 96.22 | 74.58 | 94.54 | **99.99** | **100.00** | 98.64 | 94.46 |
| Ens-R (us) | 83.69 | 90.15 | 95.39 | 73.09 | 94.06 | **99.99** | **100.00** | 98.15 | 93.23 |
| Ens-F (us) | 85.84 | **93.42** | **96.54** | 70.43 | 94.47 | 99.97 | **100.00** | **98.78** | 94.88 |

