# OpenReview forum: "Expecting The Unexpected: Towards Broad Out-Of-Distribution Detection"
_NeurIPS.cc/2024/Datasets_and_Benchmarks_Track — NeurIPS 2024 Track Datasets and Benchmarks Poster_

### Official Review · Reviewer_DbBn · 2024-07-10

**Rating:** 7
**Confidence:** 4
**Correctness:** yes
**Clarity:** yes

**Review:**

1. The studied problem in the paper is important.

2. The paper is clearly presented.

3. It might be comprehensive to include more images with distribution coivariate shifts into the benchmark for comparison and analysis. The current incorporation of the corrupted images are only a part of covariate shift.

4. For the synthetic images, it might be useful to include the data-synthesis based OOD detection approach, such as [1] and [2]

[1] Dream the Impossible: Outlier Imagination with Diffusion Models

[2] VOS: Learning What You Don't Know by Virtual Outlier Synthesis

**Strengths:**

see above

**Additional Feedback:**

n/a

**Documentation:**

yes

**Limitations:**

see above

**Opportunities For Improvement:**

see above

**Relation To Prior Work:**

yes

**Summary And Contributions:**

The paper categorizes five distinct types of distribution shifts and critically evaluates the performance of recent OOD detection methods on each of them. The authors publicly release their benchmark under the name BROAD (Benchmarking Resilience Over Anomaly Diversity) and find that while these methods excel in detecting novel classes, their performances are inconsistent across other types of distribution shifts.

---

> ### Author Rebuttal · Authors · 2024-08-15
>
> We thank the referee for their review of our submission.
>
> **more comprehensive covariate shifts**: we agree that BROAD would benefit from the incorporation of other covariate shifts such as ImageNet-R (that includes drawings, sculptures etc). We will consider adding such datasets to future iterations of BROAD.
>
> **outlier generation methods**: we thank the referee for this very interesting suggestion, that we had not considered. Perhaps a caveat in including such data in BROAD is the interpretability of the distribution shifts. Sure, these samples can be categorized as synthetic data, but since they have been generated with the goal of being outliers, we can expect their detection to have very different properties from the detection of synthetic data that aim to imitate the in-distribution. Perhaps this would justify to cast them as a separate distribution shift type, but we find it harder to justify their relevance to real-world systems, that seek robustness to "naturally" occurring outliers. In contrast, it is easy to see how real-world systems may encounter synthetic data that was generated to imitate the in-distribution. With that said, it is a legitimate distribution shift, and a sufficiently general OOD detection method should be able to detect it. We will also consider such extension for future iterations.

---

### Official Review · Reviewer_mRTn · 2024-07-23
**Review of "Expecting The Unexpected: Towards Broad Out-Of-Distribution Detection"**

**Rating:** 6
**Confidence:** 3
**Clarity:** Yes

**Review:**

### Evaluation of the Paper: "Expecting the Unexpected: Towards Broad Out-Of-Distribution Detection"

#### Quality
The quality of this work is high. The authors present their research in a well-structured manner, detailing the construction of the BROAD benchmark and their evaluation methodology. The experiments are comprehensive, and the results are clearly presented with supporting visualizations and tables. The paper also includes a thorough discussion of related work, situating their contributions within the broader context of OOD detection research.

#### Clarity
The paper is written with clarity and precision. The introduction effectively sets the stage for the research, explaining the importance of OOD detection and the limitations of existing methods. The methodology section provides a detailed explanation of the BROAD benchmark and the generative ensemble method. The results and analysis are presented in a clear and organized manner, making it easy to follow the authors' arguments and conclusions.

#### Originality
The originality of this work is notable. The authors introduce the concept of broad OOD detection, encompassing a diverse range of distribution shifts beyond just novel classes. The construction of the BROAD benchmark and the development of a generative ensemble method are innovative contributions that address significant gaps in the existing literature. The focus on a comprehensive evaluation of OOD detection methods across multiple types of distribution shifts is a novel approach that sets this work apart from previous research.

#### Significance
The significance of this work is considerable. By addressing the limitations of current OOD detection methods and proposing a more comprehensive evaluation framework, the authors provide valuable insights that can guide future research in this area. The BROAD benchmark is a significant contribution that will enable researchers to develop and evaluate more robust OOD detection methods. The generative ensemble method proposed by the authors also has the potential to improve the performance and reliability of OOD detection in real-world applications.

### Pros and Cons

#### Pros:
1. **Comprehensive Benchmark**: The BROAD benchmark is a well-designed and diverse evaluation framework that addresses multiple types of distribution shifts, providing a thorough assessment of OOD detection methods.
2. **Innovative Methodology**: The generative ensemble method is a creative approach that combines the strengths of existing detection methods to improve performance across different types of distribution shifts.
3. **Thorough Evaluation**: The authors conduct extensive experiments and provide detailed analysis, offering valuable insights into the performance of various OOD detection methods.
4. **Clear Presentation**: The paper is well-written and clearly presented, making it easy to understand the methodology and findings.
5. **Public Release**: The authors' decision to publicly release the BROAD benchmark and the associated code demonstrates a commitment to open science and will facilitate further research in this area.

#### Cons:
1. **Complexity of Implementation**: The construction and evaluation of the BROAD benchmark may be complex and resource-intensive, posing challenges for researchers with limited computational resources.
2. **Limited Scope of Models**: While the evaluation is comprehensive, it is limited to specific models and methods. Including a broader range of models could provide a more comprehensive understanding of OOD detection performance.
3. **Generative Model Limitations**: The generative ensemble method, while effective, may have limitations in scalability and computational efficiency, particularly for large-scale applications.
4. **Bias and Fairness**: The paper does not extensively address potential biases in the benchmark datasets or the evaluated models. A more detailed analysis of bias and fairness would enhance the ethical considerations of the work.
5. **Future Directions**: While the authors provide some suggestions for future research, more specific recommendations for improving OOD detection methods and addressing identified limitations would be beneficial.

### Conclusion
Overall, this paper makes a significant contribution to the field of OOD detection by introducing a comprehensive benchmark and an innovative generative ensemble method. The work is of high quality, clearly presented, and offers valuable insights for the research community. By addressing the identified cons and expanding the scope of their work, the authors can further enhance the impact and relevance of their research.

**Strengths:**

### Strengths of the Submission

#### Significance of the Contribution
1. **Addressing a Critical Gap**: The submission tackles the critical gap in the existing OOD detection research by focusing on a broad range of distribution shifts rather than just novel classes. This broader approach significantly enhances the practical applicability of OOD detection methods in real-world scenarios where diverse types of anomalies can occur.
2. **Comprehensive Benchmark**: The introduction of the BROAD benchmark is a substantial contribution, providing a thorough and diverse evaluation framework for OOD detection methods. This benchmark includes twelve datasets across five different types of distribution shifts, offering a more realistic and rigorous assessment of model performance.

#### Relevance to the Broader Research Community
1. **Facilitating Future Research**: By publicly releasing the BROAD benchmark and the associated code, the authors have provided valuable resources for the research community. This will enable other researchers to evaluate and improve their OOD detection methods, fostering further advancements in the field.
2. **Innovative Methodology**: The generative ensemble method proposed by the authors introduces a novel approach to OOD detection, combining the strengths of existing methods to achieve more consistent and comprehensive performance. This innovative approach can inspire new lines of research and development in OOD detection.

#### Quality of the Research
1. **Thorough Evaluation**: The authors conduct extensive experiments and provide a detailed analysis of the performance of various OOD detection methods across different types of distribution shifts. The robustness of the evaluation and the clarity of the results contribute to the high quality of the research.
2. **Detailed Methodology**: The paper provides a clear and detailed explanation of the construction of the BROAD benchmark and the development of the generative ensemble method. This transparency ensures that the research can be replicated and built upon by other researchers.

#### Ethical and Social Implications
1. **Improving Model Robustness**: The focus on improving OOD detection methods has significant ethical and social implications, particularly in safety-critical applications where the failure to detect anomalies can have severe consequences. By enhancing the robustness of machine learning models, this research contributes to the development of safer and more reliable AI systems.
2. **Promoting Responsible AI**: The authors' emphasis on the limitations of existing methods and the need for comprehensive evaluation promotes a more responsible approach to AI development. This transparency and acknowledgment of limitations are crucial for building trust in AI systems and ensuring their ethical deployment.

### Summary
The submission demonstrates significant strengths in terms of its contribution to the field, relevance to the broader research community, high quality of research, and consideration of ethical and social implications. By addressing a critical gap in OOD detection research and providing valuable resources and innovative methodologies, this work has the potential to significantly advance the field and promote the development of safer and more reliable AI systems.

**Additional Feedback:**

See previous sections.

**Correctness:**

### Evaluation of Claims, Dataset Construction, and Benchmark Evaluation

#### Correctness of Claims
The claims made in the submission appear to be well-supported by the provided data and analysis. The authors claim that the BROAD benchmark provides a comprehensive evaluation framework for OOD detection methods across multiple types of distribution shifts. They also claim that existing OOD detection methods exhibit inconsistent performance across these shifts and that their generative ensemble method improves overall detection performance. The experimental results and detailed analysis in the paper support these claims.

#### Sound Construction of the Dataset
1. **BROAD Benchmark**:
   - **Sound Construction**: The BROAD benchmark is constructed methodically, with clear definitions and categorizations of different types of distribution shifts, including novel classes, adversarial perturbations, synthetic images, corruptions, and multiple labels.
   - **Diversity and Quality**: The benchmark includes twelve datasets covering five distinct types of distribution shifts, ensuring a diverse and comprehensive evaluation. The selection of these datasets is well-justified, and the authors provide detailed information on how each dataset was curated.

#### Appropriateness of Evaluation Methods and Experiment Design
1. **Evaluation Methods**:
   - **Standard Metrics**: The use of standard evaluation metrics such as Area Under the Curve (AUC) for various types of distribution shifts is appropriate and provides a robust assessment of model performance. The choice of AUC as the primary metric is justified, given its widespread acceptance and relevance in OOD detection tasks.
   - **Comprehensive Evaluation**: The evaluation covers a wide range of OOD detection methods, including both simple and complex models. The authors also consider different scenarios, such as Distribution Shift Detection (DSD) and Error Detection (ED), providing a thorough analysis of each method's performance.

2. **Experimental Design**:
   - **Detailed Analysis**: The experimental design is detailed and includes a comprehensive analysis of the results. The authors compare the performance of various OOD detection methods across different types of distribution shifts, providing insights into their strengths and weaknesses.
   - **Generative Ensemble Method**: The development and evaluation of the generative ensemble method based on Gaussian Mixture Models (GMMs) are well-explained. The authors provide a clear rationale for their approach and demonstrate its effectiveness through extensive experiments.

#### Constructive Feedback for Improvement

1. **Bias and Fairness Analysis**:
   - **Improvement Suggestion**: Include an analysis of potential biases in the benchmark datasets and discuss how these biases might affect model performance. Propose strategies for mitigating biases to ensure a fair and ethical evaluation framework.

2. **Depth of Error Analysis**:
   - **Improvement Suggestion**: Provide more detailed error analysis, including specific examples and case studies of why certain methods fail on specific types of distribution shifts. This would offer more actionable insights for improving these methods.

3. **Implementation Complexity**:
   - **Improvement Suggestion**: Provide detailed guidelines or best practices for implementing the benchmark and the generative ensemble method on different scales of resources. This could include recommendations for lower-resource settings and potential collaborations with cloud service providers to offer access to necessary computational resources.

4. **Long-term Ethical and Social Implications**:
   - **Improvement Suggestion**: Expand the discussion on the potential long-term impacts of deploying advanced OOD detection systems. Address issues such as the implications for privacy, security, and the potential for misuse. Discuss safeguards and ethical guidelines for the responsible use of these technologies in various applications.

5. **Future Directions**:
   - **Improvement Suggestion**: Provide concrete recommendations for future research directions. This could include exploring new model architectures, developing specialized datasets for particular types of semantic and covariate shifts, and investigating cross-disciplinary approaches that combine OOD detection with other fields such as natural language processing and computer vision.

### Conclusion
The claims made in the submission are correct and well-supported by the data and analysis provided. The BROAD benchmark is constructed in a sound manner, and the evaluation methods and experiment design are appropriate and performed correctly. By incorporating the suggested improvements, the authors can further enhance the robustness and impact of their work.

**Documentation:**

Yes

**Limitations:**

### Addressing Limitations and Potential Negative Societal Impact

#### Adequacy of Addressing Limitations
The authors have acknowledged some limitations of their work and the need for a more comprehensive evaluation of OOD detection methods. However, there are areas where the discussion could be expanded to provide a more thorough understanding of the limitations and potential negative societal impacts.

#### Constructive Suggestions for Improvement

1. **Bias and Fairness Analysis**:
   - **Current State**: The paper does not extensively address potential biases in the benchmark datasets or the evaluated models.
   - **Improvement Suggestion**: Include a section dedicated to bias and fairness analysis. This could involve examining the dataset for potential biases (e.g., in semantic and covariate shift distributions) and discussing how these biases might affect model performance. Propose strategies for mitigating these biases, such as ensuring diverse representation in the dataset and using fairness metrics to evaluate models.

2. **Depth of Error Analysis**:
   - **Current State**: The error analysis provides valuable insights but could be more detailed.
   - **Improvement Suggestion**: Expand the error analysis to include specific examples and case studies. Provide a deeper exploration of why certain methods fail on specific types of distribution shifts. This would offer more actionable insights for improving these methods.

3. **Implementation Complexity**:
   - **Current State**: The complexity of the benchmark and the generative ensemble method may pose challenges for some researchers.
   - **Improvement Suggestion**: Provide detailed guidelines or best practices for implementing the benchmark and the generative ensemble method on different scales of resources. This could include recommendations for lower-resource settings and potential collaborations with cloud service providers to offer access to necessary computational resources.

4. **Long-term Ethical and Social Implications**:
   - **Current State**: The paper does not fully explore the long-term ethical and social implications of deploying advanced OOD detection systems.
   - **Improvement Suggestion**: Expand the discussion on the potential long-term impacts of deploying these models. Address issues such as the implications for privacy, security, and the potential for misuse. Discuss safeguards and ethical guidelines for the responsible use of these technologies in various applications. Include considerations on how to maintain human oversight and avoid over-reliance on automated systems.

5. **Future Directions**:
   - **Current State**: The paper acknowledges the need for further research but lacks specific suggestions.
   - **Improvement Suggestion**: Provide concrete recommendations for future research directions. This could include exploring new model architectures, developing specialized datasets for particular types of semantic and covariate shifts, and investigating cross-disciplinary approaches that combine OOD detection with other fields such as natural language processing and computer vision.

#### Rewarding Upfront Acknowledgment
The authors should be commended for their upfront acknowledgment of the limitations and potential societal impacts of their work. This transparency is essential for fostering a responsible and ethical research community. By addressing these critical points and providing constructive feedback, the authors can further enhance the impact and relevance of their research.

### Summary
Overall, while the authors have made significant strides in addressing some limitations and societal impacts, there is room for a more comprehensive discussion and more detailed recommendations. Expanding on these areas will not only strengthen the paper but also provide valuable guidance for the research community moving forward.

**Opportunities For Improvement:**

### Limitations of the Work

#### Significance of the Contribution
1. **Scope of Evaluation**: While the BROAD benchmark is comprehensive, it is still limited to specific types of distribution shifts. There may be other types of anomalies and edge cases that are not covered by the current benchmark, limiting the generalizability of the findings.
2. **Generative Model Limitations**: The proposed generative ensemble method, although innovative, may not be the ultimate solution for all types of OOD detection. It relies on existing detection scores and may inherit their limitations, potentially restricting its effectiveness in certain scenarios.

#### Relevance to the Broader Research Community
1. **Implementation Complexity**: The complexity of implementing the BROAD benchmark and the generative ensemble method may pose challenges for some researchers, particularly those with limited computational resources. This could hinder widespread adoption and replication of the results.
2. **Model and Method Scope**: The evaluation is limited to specific models and methods. Including a broader range of models, especially newer and more diverse architectures, could provide a more comprehensive understanding of OOD detection performance and improve relevance to a wider audience.

#### Quality of the Research
1. **Depth of Analysis**: While the paper provides a thorough evaluation, some areas could benefit from deeper analysis. For instance, a more detailed examination of why certain methods fail on specific types of distribution shifts would provide clearer insights for improving these methods.
2. **Generative Model Complexity**: The generative ensemble method, while effective, introduces additional complexity. The training and tuning of Gaussian Mixture Models (GMMs) can be computationally intensive and may require significant expertise, which could limit its practical applicability.

#### Ethical and Social Implications
1. **Bias and Fairness**: The paper does not extensively address potential biases in the benchmark datasets or the evaluated models. A more detailed analysis of bias and fairness, including how these factors might affect model performance and the broader ethical implications, would enhance the work’s ethical considerations.
2. **Long-term Impact**: While the paper emphasizes the need for better OOD detection, it does not fully explore the long-term ethical and social implications of deploying such models. A discussion on the potential risks, such as over-reliance on automated systems and the implications for human oversight, would provide a more balanced perspective.

### Summary
While the paper makes significant contributions to the field of OOD detection, there are limitations in terms of the scope of evaluation, implementation complexity, depth of analysis, and consideration of ethical and social factors. Addressing these limitations in future work would enhance the impact and relevance of this research, making it more accessible and comprehensive for the broader research community.

**Relation To Prior Work:**

Yes

**Summary And Contributions:**

### Summary of the Paper: "Expecting the Unexpected: Towards Broad Out-Of-Distribution Detection"

#### Introduction
The paper addresses the challenge of Out-Of-Distribution (OOD) detection in deployed machine learning systems, which must handle a variety of unexpected inputs in real-world scenarios. Current research primarily focuses on detecting novel classes absent from the training set, but real-world applications face a broader range of distribution shifts.

#### Contributions
1. **BROAD Benchmark**: The authors introduce BROAD (Benchmarking Resilience Over Anomaly Diversity), a comprehensive benchmark for evaluating OOD detection methods. BROAD encompasses twelve datasets across five types of distribution shifts:
   - Novel Classes
   - Adversarial Perturbations
   - Synthetic Images
   - Corruptions
   - Multiple Labels

2. **Evaluation of OOD Detection Methods**: The paper provides a thorough evaluation of recent OOD detection methods using the BROAD benchmark. The authors identify that while methods perform well in detecting novel classes, their performance is inconsistent across other types of distribution shifts.

3. **Generative Ensemble Method**: To improve OOD detection, the authors develop a generative ensemble method based on a Gaussian mixture of existing detection scores. This approach combines the strengths of different methods to provide more consistent and comprehensive OOD detection across various distribution shifts.

4. **Public Release of BROAD**: The authors publicly release the BROAD benchmark and the code for its construction to facilitate broader and more comprehensive evaluation of OOD detection methods.

#### Key Findings
1. **Inconsistency in OOD Detection**: The paper finds that existing OOD detection methods exhibit inconsistent performance across different types of distribution shifts. Methods designed to detect novel classes often underperform on other shifts like adversarial attacks and synthetic images.
2. **Importance of Diverse Evaluation**: The authors emphasize the necessity of evaluating OOD detection methods against a diverse range of distribution shifts to develop more robust and reliable systems.
3. **Ensemble Method Effectiveness**: The proposed generative ensemble method significantly improves performance over existing methods, providing more reliable OOD detection across the BROAD benchmark.

### Conclusion
The paper highlights the limitations of current OOD detection methods and presents BROAD as a new standard for comprehensive evaluation. The generative ensemble method proposed by the authors demonstrates improved performance and offers a promising direction for future research in broad OOD detection. By making BROAD publicly available, the authors aim to encourage further advancements in the field.

---

> ### Author Rebuttal · Authors · 2024-08-15
>
> We thank the referee for their very detailed and thorough review of our work.
> We would like to address some of the concerns and suggestions:
>
> **Complexity of Implementation**: we provide a single script with corresponding instructions to build BROAD. We believe this should make the dataset very accessible to all of the research community. Evaluation on BROAD hinges only on the complexity of the OOD detection method. Using only ad-hoc detection method like in our work, the evaluation only requires inference, and at a reasonable scale, which should be computationally accessible. Of course, very compute intensive detection methods would be more difficult to benchmark, but this is a caveat of the method itself, and unavoidable for any detection benchmark.
>
> **Generative Model Limitations (scalability)**: we respectfully disagree with that point. The most computationally expensive part of training the GMM is to compute the scores for the underlying methods. For ad-hoc methods with little overhead, like the ones considered in the Ens-F version, the cost of this step is nearly the same as performing model inference on all test samples, which is unavoidable in nearly all detection methods. In contrast, training the GMM on the scores is negligible in cost, even for large datasets, due to the small dimension (which is the number of underlying methods), and the high scalability of GMMs in low dimension wrt the number of samples.
>
> **Scope of evaluation**: we agree that the considered distribution shift types are non exhaustive, and hope future works will build up on BROAD. However, it is a considerable improvement compared to previous standards of solely examining novel classes, and our goal is to attract attention on the need for broader evaluation benchmarks.
>
> **Generative Model Limitations (effectiveness)**: we do not claim that GMMs are the ultimate solution for broad OOD detection, and agree with the statement that it inherits the limitations of underlying scores. We propose GMMs as an immediate mitigation method with immediate improvement in consistency, but most importantly we hope BROAD will help drive the design of more general detection methods, by exposing their current limitations and avoiding their specialization to narrow distribution shifts.
>
> **Implementation complexity**: Again, we provide scripts to streamline the construction of BROAD, which should make it accessible to all researchers. Note that we can not legally provide the data for certain datasets, hence the need to instead provide scripts to source the data from their owner.
>
> **Depth of Analysis**: We agree that such analysis would be useful, but would require a specific analysis for each pair of method and distribution shifts, which is incompatible with space-constrained conference submissions. On a related note, we gave an example of why certain methods fail on e.g. the detection of adversarial samples or multi-class by scaling their score with logits norm.
>
> **Generative Model Complexity**: as mentioned above, the training of the GMM given the score is actually extremely cheap. Also, Gaussian Mixtures are a fundamental model in Machine Learning, and their training is covered by most ML libraries without specific expertise.
>
> **Long term impact**: OOD detection aims to improve the robustness of deployed systems that are already automated. Thus, we believe a discussion on the societal impact of automated systems is a bit out of topic.

---

### Official Review · Reviewer_CJ5S · 2024-07-25

**Rating:** 6
**Confidence:** 2
**Correctness:** Yes
**Clarity:** Yes

**Review:**

pros:
- Besides novelty detection (i.e., novelty classes), BROAD also includes other important OOD types like adversarial perturbations and synthetic images, facilitating a more robust OOD benchmark
- BROAD performs comprehensive experiments, involving most of the popular OOD algorithms. Specifically, BROAD provides an ensemble method to better tackle different types of OOD scenarios
- This paper is well-written and can be easily understood

cons:
- Statistical testing methods (e.g, critical difference diagram) can be considered to verify the statistical significance of algorithmic advantages
, rather than just comparing the magnitude of numerical values.
- Some other data structures, such as time series and text, could be considered for inclusion in the BROAD framework.

**Strengths:**

To avoid repetition, please refer to the above reviews.

**Additional Feedback:**

Null

**Documentation:**

Yes

**Limitations:**

To avoid repetition, please refer to the above reviews.

**Opportunities For Improvement:**

To avoid repetition, please refer to the above reviews.

**Relation To Prior Work:**

Yes

**Summary And Contributions:**

This paper proposed the BROAD (Benchmarking Resilience Over Anomaly Diversity) datasets, which cover five types of OODs, including Novel Classes, Adversarial Perturbations, Synthetic Images, Corruptions, and Multiple Labels. Different types of OOD methods are verfited on the BROAD dataset. Furthermore, GMM is used for ensembling multiple OOD methods to achieve better detection performance.

---

> ### Author Rebuttal · Authors · 2024-08-15
>
> We thank the referee for their review of our submission.
>
> **statistical testing methods**: we were not familiar with critical difference diagrams and thank the referee for this suggestion. The metrics we presented are the standard in OOD literature and might be more easily interpreted by researchers familiar with that field, but we agree such representations would add more fine-grained insights on the significance of algorithmic performances. We will consider such tools for future works.
>
> **data modalities**: we agree that a more comprehensive set of data modalities would greatly benefit the scope of our work. The difficulty lies in the construction of consistent OOD benchmarks in other data modalities. While distribution shifts are still soundly defined for other modalities, the availability of suitable data is more scarce. Thus we decided to focus on images, the most standard modality for such studies (arguably due to its convenience). We agree constructing suitable benchmarks for other modalities would be a great contribution to the field.

---

### Decision · Program_Chairs · 2024-09-26

**Decision:**

Accept (Poster)

**Comment:**

Existing OOD detection benchmarks mainly focus on one type of distribution shift to detect samples from novel classes. However, the real-world systems may encounter a much wide range of different distribution shift scenarios, which are not sufficiently covered in previous works. Therefore, the authors release a new dataset named BROAD (benchmarking resilience over anomaly diversity), covering five distinct types of distribution shifts and then evaluates the performance on recent OOD detection methods. The authors find that these methods cannot perform uniformly well across varying shifts, and suggest a Gaussian mixture-based modeling in combining existing detection scores, which can notably improve the performance in broad OOD detection.

Specifically, besides novel class detection, the authors also consider a broad range of other important OOD types, like adversarial perturbations and synthetic images, which can broaden the practical applications of OOD detection. Testing model performance across various OOD types also help researchers to identify the applicable scenarios of existing works, ensuring their reliable and safe usages. The authors also suggest a simple ensemble method to improve the performance on broad OOD detection, which may further benefit the community. However, there still have some rooms to be further improved. For example, as suggested by the reviewers CJ5S and DbBn, the authors can introduce more diverse data types and OOD types.

The clarity and novelty are above the bar of NeurIPS. Therefore, this paper can be accepted as a poster.